# Complete Chloroplast Genome of an Endangered Species *Quercus litseoides*, and Its Comparative, Evolutionary, and Phylogenetic Study with Other *Quercus* Section *Cyclobalanopsis* Species

**DOI:** 10.3390/genes13071184

**Published:** 2022-07-01

**Authors:** Yu Li, Tian-Rui Wang, Gregor Kozlowski, Mei-Hua Liu, Li-Ta Yi, Yi-Gang Song

**Affiliations:** 1College of Forestry and Biotechnology, Zhejiang A&F University, Lin’an, Hangzhou 311300, China; liyu980625@163.com (Y.L.); mhliu@zafu.edu.cn (M.-H.L.); 2Eastern China Conservation Centre for Wild Endangered Plant Resources, Shanghai Chenshan Botanical Garden, Shanghai 201602, China; wtianrui@163.com (T.-R.W.); gregor.kozlowski@unifr.ch (G.K.); 3Department of Biology and Botanic Garden, University of Fribourg, Chemin du Musée 10, 1700 Fribourg, Switzerland; 4Natural History Museum Fribourg, Chemin du Musée 6, 1700 Fribourg, Switzerland

**Keywords:** Fagaceae, montane cloud forests, *Cyclobalanopsis litseoides*, repeat sequences, plastome

## Abstract

*Quercus litseoides*, an endangered montane cloud forest species, is endemic to southern China. To understand the genomic features, phylogenetic relationships, and molecular evolution of *Q. litseoides*, the complete chloroplast (cp) genome was analyzed and compared in *Quercus* section *Cyclobalanopsis*. The cp genome of *Q. litseoides* was 160,782 bp in length, with an overall guanine and cytosine (GC) content of 36.9%. It contained 131 genes, including 86 protein-coding genes, eight ribosomal RNA genes, and 37 transfer RNA genes. A total of 165 simple sequence repeats (SSRs) and 48 long sequence repeats with A/T bias were identified in the *Q. litseoides* cp genome, which were mainly distributed in the large single copy region (LSC) and intergenic spacer regions. The *Q. litseoides* cp genome was similar in size, gene composition, and linearity of the structural region to those of *Quercus* species. The non-coding regions were more divergent than the coding regions, and the LSC region and small single copy region (SSC) were more divergent than the inverted repeat regions (IRs). Among the 13 divergent regions, 11 were in the LSC region, and only two were in the SSC region. Moreover, the coding sequence (CDS) of the six protein-coding genes (*rps12*, *matK*, *atpF*, *rpoC2*, *rpoC1*, and *ndhK*) were subjected to positive selection pressure when pairwise comparison of 16 species of *Quercus* section *Cyclobalanopsis*. A close relationship between *Q. litseoides* and *Quercus edithiae* was found in the phylogenetic analysis of cp genomes. Our study provided highly effective molecular markers for subsequent phylogenetic analysis, species identification, and biogeographic analysis of *Quercus*.

## 1. Introduction

Trees provide habitat for half the world’s known terrestrial plant and animal species and are highly significant components of biodiversity and carbon storage in many ecosystems [1]. Recently, the vital importance of trees has received increasing attention ecologically, culturally, and economically [2,3]. Globally, more than 30% of tree species were classified as threatened in the State of the World’s Trees report [1].

*Quercus* (oaks) is the largest genus (ca. 430 species) in Fagaceae and is one of the most important, species-rich, and entirely woody eudicot families [2]. There are two subgenera (*Quercus* and *Cerris*) with eight sections in the genus [4]. The taxonomy, in which section *Cyclobalanopsis* is nested in *Quercus*, has been largely investigated for their phylogenetic relationships [5,6,7]. It has been treated as a separate genus in Flora of China [8]. *Quercus* is predominantly found in the temperate and subtropical forest ecosystems of the Northern Hemisphere. According to the IUCN’s method for calculating threatened proportions incorporating data deficient species, 31% of oaks are threatened with extinction [9,10,11].

*Quercus litseoides* Dunn, an evergreen vulnerable tree/shrub species, forms small fragments of pure forests or sparse forests near the top of mountains with elevations between 700 and 1000 m [8]. It belongs to the *Quercus* section *Cyclobalanopsis*, with only five known populations endemic to South Guangdong and Hong Kong, China [8,12]. Thus, this species showed an island distribution pattern that is known as “sky islands”. The limited populations of *Q. litseoides* are especially threatened by habitat destruction, soil erosion, and climate change [9,13]. Assessment of rough data has finally led to *Q. litseoides* being classified as vulnerable [9].

Morphologically, the obovate-oblanceolate to narrowly elliptic leaf blade of *Q. litseoides* resembles another sky island species, *Quercus arbutifolia*. Leaf epidermal features of *Q. litseoides* (uniseriate trichome with single-celled trichome base (STB)) showed that it systematically belongs to the STB group [14]. The molecular phylogenetic work on *Quercus* always lacks the *Q. litseoides* species, owing to its limited distribution and the difficulty in acquiring materials [5,6,7]. Thus, few studies focused on its phylogenetic and phylogenomic analysis, and genetic and structural diversity, and so on.

The chloroplast (cp) is an essential maternal hereditary organelle in green plant cells with an independent circular genome and plays a critical role in photosynthesis and carbon fixation [15,16,17]. Owing to its conserved genome structure, gene composition, and variation ratio, the cp genome is used for comparative genomics, species identification, plant evolutionary studies, tracking seed dispersal, and studying the structural diversity and evolution of organellar genomes [7,18,19,20].

Owing to the benefits of next-generation sequencing technologies, whole cp genome sequencing is affordable and more efficient than ever before. Fifty complete cp genomes of *Quercus* have been published in the National Center for Biotechnology Information (NCBI) database, including 14 from the *Quercus* section *Cyclobalanopsis*. Thus, we have opportunities to (1) elaborate on the typical structural characteristics of *Q. litseoides*, (2) examine abundant, simple sequence repeats (SSRs) and repeat structures in the whole cp genome of *Q. litseoides* to provide markers for phylogenetic and genetic studies, (3) identify evolutionary selection pressure of the coding sequence (CDS) of the 78 shared protein-coding genes (PCGs), and (4) explore the phylogenomic position of *Q. litseoides*.

## 2. Materials and Methods

### 2.1. Plant Material, DNA Extraction, and Sequencing

Fresh, healthy leaf samples of *Q. litseoides* were collected from the Wu-Tong Mountain in Shenzhen, Guangdong Province (113°17′ E, 22°23′ N; Alt. 944 m). The leaves were desiccated in silica gel and deposited at the herbarium of the Shanghai Chenshan Botanical Garden. Total genomic DNA was extracted and purified from leaf tissues using a modified cetyl trimethyl ammonium bromide (CTAB) protocol [21]. DNA was fragmented by ultrasonic mechanical interruption, and the DNA fragments were purified. A DNA library with an average insert size of 350 bp was constructed using the whole genome shotgun strategy, and the quantified DNA was then double-terminally sequenced based on the Illumina NovaSeq6000 (Illumina, San Diego, CA, USA) technology platform in accordance with the manufacturer’s manual at Wuhan Benagen Technology Co., Ltd. (Wuhan, China) [22].

### 2.2. Genome Assembly and Annotations

Raw reads from the sample produced at least five Gb with 150 bp pair-end read lengths by base calling analysis [23]. Clean reads were obtained by filtering low-quality sequences (quality value of Q ≤ 5 and N bases > 5%) using SOAPnuke Toolkit v.1.3.0 [24]. Genome assembly was performed using SPAdes v.3.13.0, with default parameters [25]. Prediction of coding genes and non-coding RNA annotations (ribosomal RNA (rRNA) and transfer RNA (tRNA)) was performed using CPGAVAS2 [26]. The fully annotated cp genome of the circular diagram was drawn using the online program Organellar GenomeDRAW (OGDRAW) [27].

### 2.3. Repeated Sequence Analysis

SSRs loci were identified using the online program MIcroSAtellite (MISA) [28]. The minimum number of SSRs was set to ten for mono-nucleotide; four for di- and tri-nucleotide; three for tetra-, penta-, and hexa-nucleotide SSR motifs. Composite microsatellites were identified by setting the minimum distance between the two SSRs to be less than 100 bp. Minisatellite sequence repeats (M) of at least 10 bp in length were identified using the program Tandem Repeats Finder (TRF) [29]. The alignment parameters for match, mismatch, and indels were set to be two, seven, and seven, respectively. The minimum alignment score and maximum period size were set to 80 and 500, respectively. Additionally, REPuter was applied to predict forward repeat sequences (F), reverse repeat sequences (R), complementary repeat sequences (C), and palindromic repeat sequences (P) using the following settings: minimum repeat sequence of 30 bp, Hamming distance of three, and sequence identity between the two repeats of more than 90% [30,31].

### 2.4. Genome Structure Comparisons and Sequence Divergence Analysis

The boundaries and gene rearrangement between the large single copy (LSC), small single copy (SSC), and inverted repeat (IR) regions among the 16 species (Section *Cyclobalanopsis*) were horizontally visualized using the online tool IRscope [32]. Using the cp genome of *Q. litseoides* as the reference sequence, the program mVISTA was used to identify interspecific variations across the complete cp genomes of 16 species of section *Cyclobalanopsis* in Shuffle-LAGAN mode [33,34]. The cp genome sequences of the *Quercus* section *Cyclobalanopsis* were aligned using MAFFT v.7.847 [35]. After manual adjustment using BioEdit v.7.2.5 [36], single nucleotide polymorphisms and indel sites were counted using DnaSP v.6.12.03 [37]. A sliding window analysis was further conducted to calculate hotspots of nucleotide variability (Pi) values between cp genomes following a window length of 600 base pairs and a step size of 200 base pairs [38].

### 2.5. Evolutionary Selection Pressure Analysis

To identify the evolutionary selection pressure in the cp genomes of the *Quercus* section *Cyclobalanopsis* [39], the CDS of the 78 shared PCGs of *Cyclobalanopsis* were extracted and aligned using MAFFT v.7.847. The synonymous substitution rate (Ks), nonsynonymous substitution rate (Ka), and Ka/Ks ratio (ω) were calculated using DnaSP v.6.12.03. This value made sense if Ks is not equal to zero. Based on Ka/Ks ratio, the evolutionary selection of CDS was classified as positive selection (Ka/Ks > 1), neutral selection (Ka/Ks = 1), or purifying selection (Ka/Ks < 1).

### 2.6. Phylogenetic Analyses

Thirty-four complete cp genome sequences, comprising one new cp genome sequence and 33 cp genome sequences of five sections of *Quercus* species from the GenBank database, were used to reconstruct phylogenetic relationships. *Trigonobalanus doichangensis*, *Fagus engleriana*, and *Juglans mandshurica* were used as outgroup species. The GenBank accession numbers for each taxon used are shown in Appendix A. Homblock v1.0 was used to first screen out homologous sequences of the whole cp genomes of these 37 species [40], and the online software Circoletto was then used to visualize the alignment sequence of *Q. litseoides* and the homologous sequences of all species [41]. Next, the phylogenetic trees were reconstructed using two methods: Maximum Likelihood (ML) and Bayesian Inference (BI), using IQtree v.1.6.12 and MrBayes v3.2.7 [42,43], respectively. The ML tree adopted TVM + F + R2 as the best nucleotide replacement model with 1000 bootstrap replicates. The BI tree was set as follows: Markov chain Monte Carlo simulations (MCMC) algorithm for 5,000,000 generations with four incrementally heated chains, starting from random trees, and sampling one out of every 100 generations. The first 25% of trees were discarded as burn-in. The constructed phylogenetic trees were further edited and visualized using FigTree v.1.4.4 (http://tree.bio.ed.ac.uk/software/fifigtree/) (accessed on 29 May 2022).

## 3. Results

### 3.1. Chloroplast Genome Assembly and Annotation of Q. litseoides

Five Gb clean reads were generated in total from the genomic DNA of *Q. litseoides* using the Illumina sequencing. The complete cp genome of *Q. litseoides* has a quadripartite structure comprising 160,782 bp, including an LSC region of 90,235 bp and an SSC region of 18,867 bp, which were separated by a pair of IR regions (IRa and IRb) of 25,840 bp (Table 1 and Figure 1). The overall guanine and cytosine (GC) content of the cp genome of *Q. litseoides* was 36.90%, and the corresponding values in the LSC, SSC, and IR regions were 34.74%, 31.13%, and 42.77%, respectively (Table 1).

A total of 131 predicted genes in the cp genome of *Q. litseoides* were assigned to three groups based on their functions: 86 PCGs, 37 tRNA genes, and eight rRNA genes (Table 1). The GC content of all the genes was 39.5%, with 37.88% for PCGs, 53.2% for tRNA genes, and 55.49% for rRNA genes (Table 1). There were 83 genes (61 PCGs and 22 tRNA genes) in the LSC region, 12 genes (11 PCGs and one tRNA gene) in the SSC region, and 36 genes (seven PCGs, seven tRNA, and four rRNA genes duplicated) in the IR regions (Figure 1 and Table 1). In addition, the *rps12* and *ycf1* genes span two regions between IRs and LSC/SSC, respectively. Moreover, the *rps12* gene was recognized as a trans-spliced gene, with exons located in the IR and LSC regions (Figure 1).

Based on the cp genome annotation of *Q. litseoides*, 113 unique genes were divided into four functional categories with 18 groups. Among the 113 unique genes, there were 60 genes related to transcription and translation, 44 genes related to photosynthesis, five genes related to biosynthesis, and four genes whose functions were unknown. A total of 18 genes in the cp genome of *Q. liteoides* contained introns (12 PCGs and six tRNA genes), of which 15 genes (*trnK-UUU*, *trnG-GCC*, *trnL-UAA*, *trnV-UAC*, *trnI-GAU*, *trnA-UGC*, *rps16*, *rpl16*, *rpl2*, *rpoC1*, *atpF*, *ndhA*, *ndhB*, *petB*, and *petD*) contained one intron, whereas the other three genes (*ycf3*, *clpP*, and *rps12*) contained two introns (Table 2).

### 3.2. Repeat Sequences in the Chloroplast Genome of Q. litseoides

The SSRs, minisatellite sequences, dispersed repeat sequences, and palindromic repeat sequences were analyzed in the cp genome of *Q. litseoides*. A total of 165 SSRs were classified into five types (mono-, di-, tri-, tetra-, penta-nucleotide repeats). The first two types (mono- and di-nucleotide repeats) accounted for 87.28% of SSRs, and the proportions of the other three types were 4.25%, 6.67%, and 1.82%, respectively. It is worth noting that 42 composite microsatellites were identified because the minimum distance between the two SSRs was less than 100 bp. Most SSRs were composed of A and T, indicating a strong A/T bias. At the same time, the distribution of SSRs in the LSC region (66.7%) was higher than that in the IR (20.6%) and SSC regions (12.7%). More than 64.8% of the SSRs were located in the intergenic spacer regions (IGS) and 35.2% in the gene regions (Table 3).

For the long repeat sequences, we detected seven M, 14 F, four R, two C, and 21 P. The length of the repeat units ranged from 19 to 56 bp (mainly between 30 and 40 bp), and the repeat sequences had two to four repeats. Most of the long repeat sequences were distributed in the LSC region, especially all reverse and complementary repeat sequences. Six of the seven M were located in IR regions. Meanwhile, there were four sequences distributed in the regions between LSC and IRs, four between SSC and IRs, and seven between IRa and IRb. In contrast, 26 sequences belonged to the gene regions, while the others were located in the intergenic spacer regions. The long repeat sequences were mainly distributed in the following gene regions: *trnG-GCC*, *trnS-GGA*, *trnS-UGA*, *rpl2*, *rpl12*, *psaB*, *ndhA*, *clpP*, *ycf1*, *ycf2*, and *ycf3* (Table 4, Appendix A).

### 3.3. Genome Structure Comparisons and Sequence Divergence of Quercus Section Cyclobalanopsis

The length of the cp genome in section *Cyclobalanopsis* changed little, with 445 bp, ranging from 160,533 bp (*Quercus acuta*) to 160,978 bp (*Quercus edithiae*). All cp genomes in section *Cyclobalanopsis* were shorter than those in the other sections of *Quercus*. The length of the IR regions among different species in this section, *Cyclobalanopsis*, varied by 31 bp (Appendix A).

The junction region between LSC and IRb (JLB) lies in the IGS between *rps19* and *rpl2* genes, and the junction region between LSC and IRa (JLA) is located between the *rpl2* and *trnH* genes. Most of the section *Cyclobalanopsis* species had 11 bp shifted away from the boundary for *rps19* gene in JLB, except *Q. edithiae* and *Quercus multinervis*, which had one bp shift. The *trnH* gene has 14 to 16 bp shifted from JLA. The *ycf1* gene crossed the junction regions between IRa/IRb and SSC (located in JSA and JSB), with the exception of the JSB of *Quercus chungii*, *Quercus acuta*, *Quercus saravanensis*, *Quercus schottkyana*, and *Q**uercus multinervis*. The *ycf1* gene (located in JSA) has 1045–1070 bp in the IRa region and 4612–4628 bp in the SSC region. However, the *ycf1* gene (located in JSB) has 1045–1060 bp in the IRb region and only 58–68 bp in the SSC region. The *ndhF* gene, located in the SSC region just beside the JSB, has a one or 11 bp shift in the species without *ycf1* gene (Figure 2).

To further investigate the divergence of cp genomes among related species, the evolution of cp genomes was explored in the *Quercus* section *Cyclobalanopsis* using the annotated cp genome of *Q. litseoides*. No structural rearrangement with high sequence similarity occurred in this section. However, the non-coding regions were more divergent than the coding regions, and the LSC and SSC regions were more divergent than the IR regions (Appendix A). Furthermore, the *ycf1*, *ndhF*, *rpl32*, and *psbD* genes in the gene regions and *petN—psbM*, *trnK-UUU—rps16*, *rps16—trnQ-UUG*, *psbM—trnD-GUC*, *psbZ— trnG-UCC*, *trnT-GGU—psbD*, *rbcL—accD*, and *rpl32—trnL-UAG* in the intergenic spacer regions were quite mutable (Appendix A).

A total of 482 variation polymorphism sites, including 335 single nucleotide polymorphism sites, 147 parsimony-informative sites, and 200 indel events, were detected by nucleotide polymorphism analysis. Nucleotide polymorphism in the IR regions of the cp genome was significantly lower than that in the LSC and SSC regions. The Pi value of nucleotide diversity in the cp genomes of section *Cyclobalanopsis* ranged from 0 to 0.01808, with an average of 0.00059. Furthermore, the analysis detected 13 highly divergent regions (Pi > 0.002), of which eight were located in the gene regions (*trnH-GUG*, *trnC-GCA*, *trnS-UGA*, *ycf1*, *ycf3*, *psaI*, *psbJ*, and *rpl22*) and five in the intergenic spacer regions (*trnK-UUU—rps16*, *rps16— trnQ-UUG*, *psbM—trnD-GUC*, *rbcL—accD*, and *ndhF—rpl32*). Among the 13 divergent regions, 11 were in the LSC region, and only two were in the SSC region (Figure 3).

### 3.4. Selective Pressure Analysis

To investigate the evolutionary characteristics of *Quercus* section *Cyclobalanopsis* in the cp genome, Ka, Ks, and the Ka/Ks (ω) ratio were calculated for the CDS of the 78 shared PCGs in the 16 cp genomes. The results showed that the Ka values ranged from 0 to 0.06019, the Ks values ranged from 0 to 0.08333, and the Ka/Ks (ω) values of 37 CDS of the PCGs were significant (Ks > 0, *p < 0.05*, Figure 4, Appendix A). The Ka/Ks (ω) values of six CDS of the PCGs (*rps12*, *matK*, *atpF*, *rpoC2*, *rpoC1*, and *ndhK*), which were distributed in the LSC region, were greater than 1, indicating that these genes had undergone positive selection. The Ka/Ks (ω) values of the other 31 CDS of the PCGs were all less than 1, suggesting that genes were under purifying selection (Figure 4, Appendix A).

### 3.5. Phylogenetic Analyses

Homologous sequences were screened for the cp genomes of 34 *Quercus* species in five sections and three outgroups (*Trigonobalanus doichangensis*, *Fagus engleriana*, and *Juglans mandshurica*). Thirty-seven homologous sequences with a length of 88,044 bp were generated. The alignment results included almost all genes (Appendix A).

The ML tree and BI tree reconstructed by homologous sequences showed a similar topological structure. All nodes of the phylogenetic trees were supported by 54–100% bootstrap values in ML analysis and 0.67–1.00 Bayesian posterior probabilities in BI analysis (Figure 5 and Appendix A). The results showed that the section *Quercus* and section *Lobatae* formed one clade. *Quercus* section *Ilex* split into two strongly supported clusters; one clade is the species distributed in the Tibetan area (*Quercus spinosa* and *Q. aquifolioides*), whereas the species from East and Central China together with *Quercus* section *Cerris* formed another clade. The bootstrap support in *Quercus* section *Cyclobalanopsis* is not so high in several nodes of the middle part. Up to now, *Quercus* section *Cyclobalanopsis* could be split into five clades. Firstly, *Q. delavayi* and *Q. acuta* diverged from the section *Cyclobalanopsis*, forming two separate clades. *Q. ningangensis* and *Q. saravanensis* were sister to the *Q. schottkyana*, constituting two clades along with other subtropical species. The montane cloud forest species (*Q. arbutifolia* and *Q. litseoides*), tropical species together with the widespread species (*Q. glauca* and *Q. multinervis*) formed the last clade (Figure 5).

## 4. Discussion

### 4.1. Architecture of cp Genomes in Quercus Section Cyclobalanopsis

In this study, we present the complete cp genome of *Q. litseoides*. Combined with the 15 related species reported previously, we performed a comparative analysis of the genomic features of *Quercus* section *Cyclobalanopsis*. Angiosperm cp genomes have a highly conserved structure and gene content [16,44]. The cp genomes of *Quercus* are highly conserved in size, quadruple structure, and GC content. There were slight differences in the total numbers of genes and unique genes in *Quercus*. Most *Quercus* species have 86 total PCGs and 79 unique PCGs, except *Q**uercus rubra*, *Q**uercus fabri*, *Q**uercus acutissima*, and *Q**. edithiae*, which have one extra gene *ycf15* [7,45,46,47]. Compared with other *Quercus* species, three tRNA genes (*trnP-GGG*, *trnT-GGU*, and *trnM-CAU*) are missing in *Q. litseoides* [7]. It has also been observed to be missing of *ycf15* and *trnP-GGG* in the cp genomes of some angiosperms [48,49]. The two copy of *trnT-GGU* and *trnM-CAU* genes in most *Quercus* species were caused by annotation and the overlap of these two genes. Most other angiosperms, such as *Musa*, *Oryza*, *Stauntonia*, and *Carya*, only recognized one copy of these two genes in the LSC region [50,51,52,53].

Repeat sequences are considered to play an important role in cp genome rearrangement and sequence differentiation [54,55]. A strong A/T bias, LSC concentration (66.7%), and IGS concentration (64.8%) for SSRs, similar to other angiosperm cp genomes, were detected in *Q. litseoides* [56,57,58]. The number and types of SSRs varied extensively when compared to those of other cp genomes in *Quercus*. The number of SSRs in *Q. litseoides* was higher than that in the other *Quercus* species, whereas fewer SSRs were distributed in the LSC and IGS regions [46,58,59,60,61,62,63]. These variations support the idea that SSRs can be used as lineage-specific markers for genetic diversity analysis and can be used as markers to understand evolutionary history [64]. The large variation in long repeats in closely related species may reflect a certain degree of evolutionary flexibility [65]. Forty-eight long repeats were detected, and several repeats occurred in the same genes (four in *ycf1*, four in *ycf**3*, six in *ycf2*, and four in *rpl12*).

The variable boundary regions are believed to be the arriving force for the variation in the angiosperm cp genomes [66,67,68,69]. The IR regions are very important in stabilizing the structure of the cp genome and are mainly responsible for variations in the length of the plastome [70]. The genes distributed in the IR regions of *Q. litseoides* in this study were similar to those of most other species, with little difference in the distribution of boundary genes. There was no obvious expansion or contraction in the IR regions of *Q. litseoides*. For the *Quercus* section *Cyclobalanopsis*, shifts of less than 16 bp were detected in JLB and JLA. The *ycf1* gene crossed the boundary of the IR and SSC regions, which had more variation in length. As expected, the relatively trivial cp genome length and variations in the boundary demonstrated the conservation of *Quercus* section *Cyclobalanopsis* plastomes. Previous studies on land plants also found that the expansion and contraction of the IR regions caused the mutual transfer of genes between the SC and IR regions or the increase and decrease of genes [71].

As the IR regions contain conserved rRNA genes with lower variability, the LSC and SSC regions had a higher level of sequence diversity than the IR regions in the *Quercus* section *Cyclobalanopsis* similar to other cp genomes [20,58,72]. The existence of a copy-dependent repair mechanism in many types of plants leads to a low replacement rate in the IR regions [72]. These mutated regions are more prone to nucleotide substitution during evolution, providing efficient molecular markers for subsequent species identification and useful data and phylogenetic information for genetic evolution analysis. In total, seven regions with higher Pi values (>0.015) were detected in *Quercus* [7], whereas only one was detected in the section *Cyclobalanopsis* (Figure 3). Compared with that of the other sections of *Quercus*, section *Cyclobalanopsis* had the lowest nucleotide diversity.

Nucleotide substitution is the driving force behind genomic evolution. Non-synonymous substitutions may alter protein function such that natural selection tends to remove these deleterious mutations, causing most species to be under negative selection pressure [73,74]. Analysis of adaptive evolution contributes to a profound understanding of gene variation, changes in protein structure and function, and the evolutionary history of species [75]. In this study, six PCGs, *rps12*, *matK*, *atpF*, *rpoC2*, *rpoC1*, and *ndhK*, were positively selected, providing evidence of the adaptive evolution of proteins. Genes with different functions have evolved at different rates [76]. The adaptive evolution of *matK*, which is involved in biosynthesis, has been detected by positive selection in multiple angiosperms [77,78]. The *atpF* and *ndhK* genes are associated with photosynthesis, whereas *rps12*, *rpoC2*, and *rpoC1* genes are associated with transcription and translation. Genes of the genetic and photosynthetic systems play an important role in the adaptation of angiosperms to terrestrial ecological environments [50,79,80]. *Q**uercus litseoides* live at high altitudes and must adapt to high UV radiation intensity, hypoxia, low temperature, and drought stress. These genes may be an important genetic basis for evolutionary adaptation at the chloroplast level.

### 4.2. Phylogeny of Chloroplast Genome of Quercus

The phylogenetic study of *Quercus* is a great challenge because of its species richness, wide distribution, and serious hybridization introgression [6,7,10]. Our cp genome phylogenetic tree supports that sections *Quercus* and *Lobatae* formed one lineage (belonging to subgenus *Quercus*), and sections *Cyclobalanopsis*, *Ilex*, and *Cerris* formed another lineage (subgenus *Cerris*) [6]. Based on the RAD-seq data, section *Ilex* is a monophyletic lineage, whereas the phylogenetic reconstruction of the cp genome was paraphyletic. The section *Cerris* formed one lineage and nested into section *Ilex* in this study, which is also different from the phylogeny based on RAD-seq data [6].

*Quercus litseoides* has the closest relationship with *Q. edithiae*, which has an overlapping geographic area. Based on the current samples, the cp genome phylogeny of section *Cyclobalanopsis* is very different from that based on RAD-seq data. When constructing phylogenetic trees based on RAD-seq data, the *Quercus* section *Cyclobalanopsis* was divided into two clades: one defined by compound trichome bases (CTB lineage) and one by single-celled trichome bases (STB lineage). However, the two lineages and relationships among species were not supported by the cp genome phylogeny in this study [5]. With the development of next-generation sequencing technologies, we could add more taxa and samples to explore and compare the phylogenomics of the cp and nuclear genomes of *Quercus* section *Cyclobalanopsis*.

## Figures and Tables

**Figure 1 genes-13-01184-f001:**
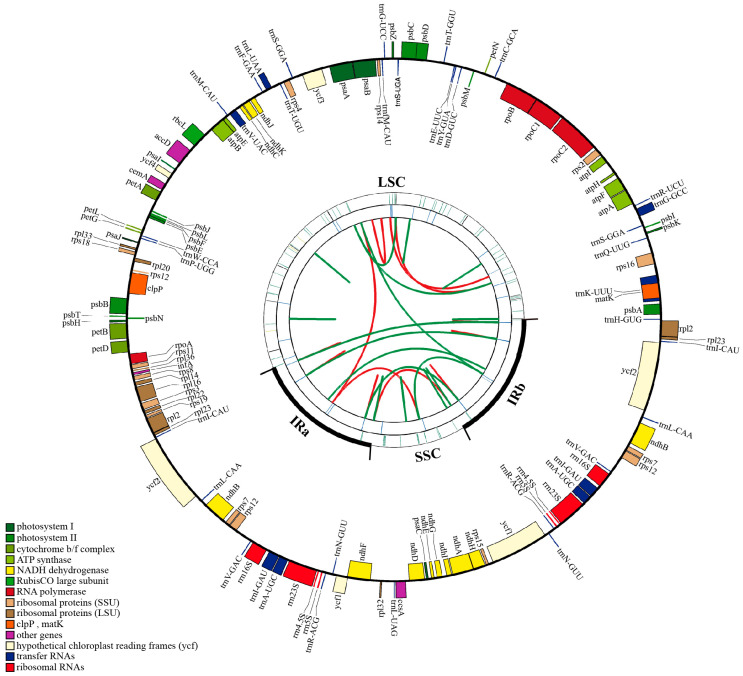
Gene map of the chloroplast genome of *Q. litseoides*. The chloroplast genome map has four circles. Outward from the center, the first circle shows forward and reverse repeats connected by red and green arcs, respectively. The second circle shows tandem repeats marked with a short bar. The third circle is the SSRs identified by MISA. The fourth circle is drawn with drawgenemap to display the gene structure on the chloroplast genome. The genes shown outside of the circle are transcribed clockwise, while those inside of the circle are transcribed counterclockwise. Genes with different functional groups are identified by different colors.

**Figure 2 genes-13-01184-f002:**
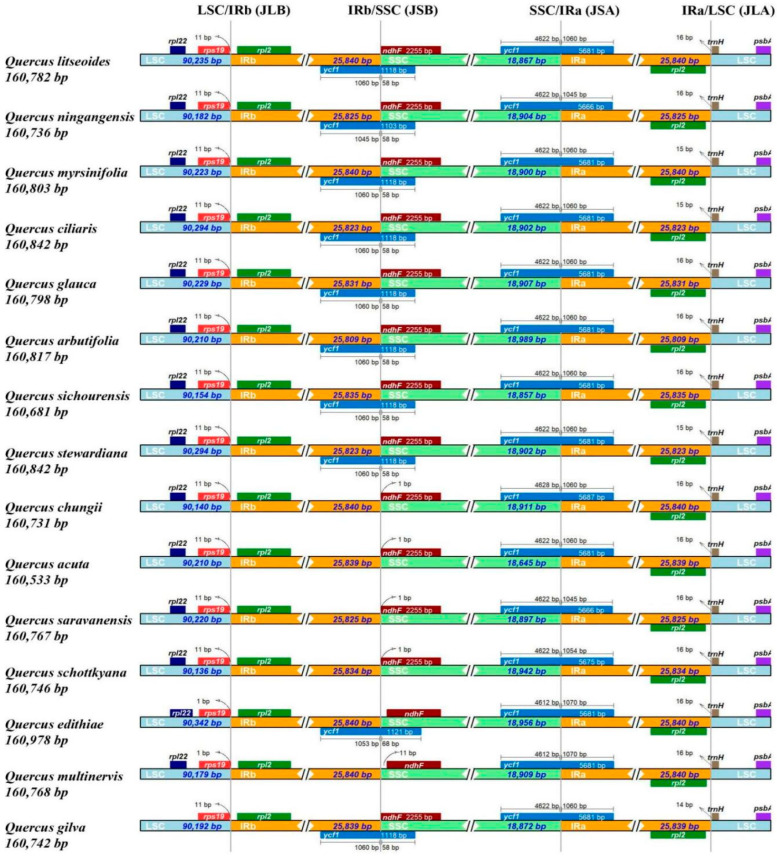
Comparison of the junction regions (JLA, JLB, JSB, JSA) of the chloroplast genomes of *Quercus* section *Cyclobalanopsis*.

**Figure 3 genes-13-01184-f003:**
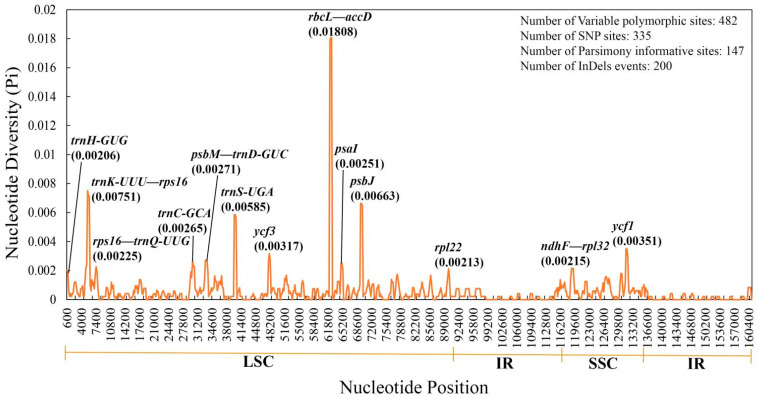
Sliding window analysis of 16 chloroplast genomes of *Quercus* section *Cyclobalanopsis*. The *x*-axis represents the site positions of the middle point of the window, and the *y*-axis represents the value of nucleotide diversity (Pi) per window.

**Figure 4 genes-13-01184-f004:**
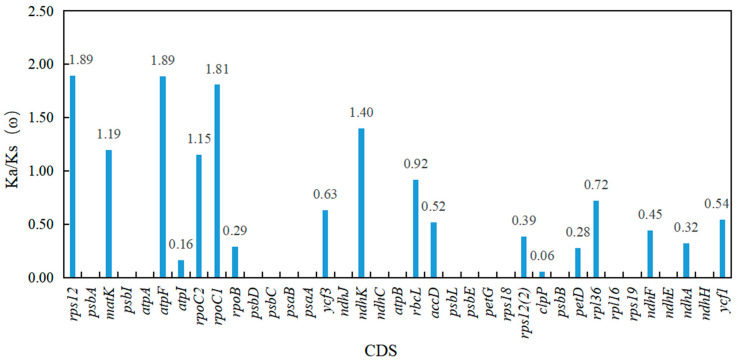
The Ka/Ks (ω) values of 37 shared functional protein-coding genes of 16 chloroplast genomes of *Quercus* section *Cyclobalanopsis*.

**Figure 5 genes-13-01184-f005:**
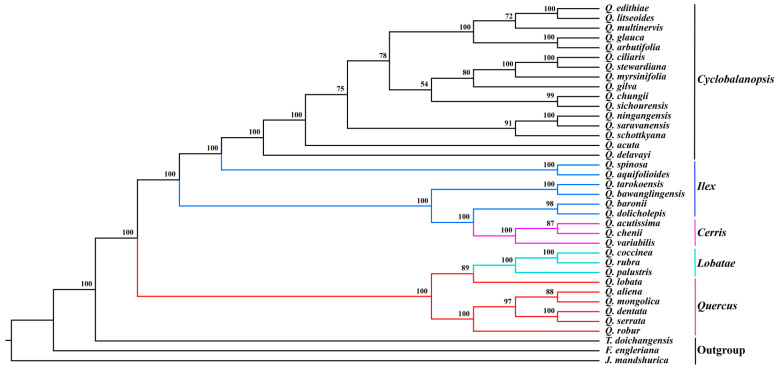
The phylogenetic tree among 37 chloroplast genome homologous sequences is based on the ML method. Values besides the branch represented bootstrap support (BS). Abbreviations: *Quercus* (*Q.*), *Trigonobalanus* (*T.*), *Fagus* (*F.*), and *Juglans* (*J.*).

**Table 1 genes-13-01184-t001:** Chloroplast genome structure and feature of *Q. litseoides*. Abbreviations: LSC (Large Single Copy), SSC (Small Single Copy), IR (Inverted Repeat), PCGs (protein-coding genes), tRNA (Transfer RNA genes), and rRNA (Ribosomal RNA genes).

Genome Feature		Length (bp)/Numbers	GC Content (%)
Structure length	Total	160,782	36.9
	LSC region	90,235	34.74
	SSC region	18,867	31.13
	IR (a/b) region	25,840	42.77
Gene numbers of different categories	Genes	131	39.5
	PCGs	86	37.88
	tRNA	37	53.2
	rRNA	8	55.49
Gene numbers of different regions	LSC region	61 (PCGs) and 22 (tRNA)	No information
	SSC region	11 (PCGs) and 1 (tRNA)	No information
	IR regions	14 (PCGs), 14 (tRNA) and 8 (rRNA)	No information

**Table 2 genes-13-01184-t002:** Genetic classification of the chloroplast genome of *Q. litseoides*. Genes marked with the * or ** sign are the gene with single or double introns, respectively. The duplicated genes located in IR regions were marked as (×2).

Category	Group	Name
Transcription and translation	Translational initiation factor	*infA*
	Ribosomal RNAs	*rrn16S* (×2), *rrn4.5S* (×2), *rrn23S* (×2), *rrn5S* (×2)
	Transfer RNAs	*trnR-UCU*, *trnfM-CAU*, *trnD-GUC*, *trnH-GUG*, *trnM-CAU*, *trnE-UUC*, *trnS-GCU*, *trnF-GAA*, *trnP-UGG*, *trnT-UGU*, *trnG-UCC*, *trnQ-UUG*, *trnY-GUA*, *trnW-CCA*, *trnS-UGA*, *trnC-GCA*, *trnT-GGU*, *trnL-UAG*, *trnS-GGA*, *trnK-UUU* *, *trnV-UAC* *, *trnL-UAA* *, *trnG-GCC* *, *trnA-UGC* *(×2), *trnI-GAU* *(×2), *trnL-CAA* (×2), *trnI-CAU* (×2), *trnN-GUU* (×2), *trnV-GAC* (×2), *trnR-ACG* (×2)
	Small subunit of ribosome (SSU)	*rps2*, *rps11*, *rps19*, *rps14*, *rps4*, *rps15*, *rps16**, *rps8*, *rps18*, *rps3*, *rps12* **(×2), *rps7* (×2)
	Large subunit of ribosome (LSU)	*rpl14*, *rpl20*, *rpl36*, *rpl33*, *rpl16* *, *rpl32*, *rpl22*, *rpl2* *(×2), *rpl23* (×2)
	DNA-dependent RNA polymerase	*rpoC2*, *rpoB*, *rpoC1* *, *rpoA*
Photosynthesis	Photosystem I	*psaB*, *psaJ*, *psaA*, *psaI*, *psaC*
	Photosystem II	*psbA*, *psbC*, *psbH*, *psbZ*, *psbI*, *psbJ*, *psbK*, *psbF*, *psbD*, *psbT*, *psbN*, *psbL*, *psbM*, *psbE*, *psbB*
	Subunit of cytochrome	*petB* *, *petN*, *petL*, *petG*, *petD* *, *petA*
	ATP synthase	*atpA*, *atpI*, *atpB*, *atpE*, *atpF* *, *atpH*
	RubisCO large subunit	*rbcL*
	NADH dehydrogenase	*ndhG*, *ndhD*, *ndhE*, *ndhK*, *ndhH*, *ndhI*, *ndhF*, *ndhA* *, *ndhJ*, *ndhC*, *ndhB* *(×2)
Biosynthesis	Maturase	*matK*
	ATP-dependent Protease	*clpP* **
	Acetyl-CoA-carboxylase	*accD*
	Envelop membrane protein	*cemA*
	C-Type cytochrome synthesis	*ccsA*
Unknown	Hypothetical chloroplast reading frames(*ycf*)	*ycf4*, *ycf3* **, *ycf1* (×2), *ycf2* (×2)

**Table 3 genes-13-01184-t003:** Simple sequence repeats (SSRs) number in the chloroplast genome of *Q. litseoides*. Abbreviations: LSC (Large Single Copy), SSC (Small Single Copy), IRs (Inverted Repeats), IGS (Intergenic Spacer Regions), GR (Gene Regions).

Repeat Type	Repeat Unit	Number (Proportion) of SSRs	Region			Location	
			LSC	SSC	IRs	IGS	GR
Mononucleotides	A/T	77 (46.67%)	60	11	6	59	18
	C/G	5 (3.03%)	5	0	0	3	2
Dinucleotides	AG/CT	19 (11.52%)	2	1	16	5	14
	AT/AT	43 (26.06%)	29	4	10	28	15
Trinucleotides	AAG/CTT	1 (0.61%)	0	1	0	0	1
	AAT/ATT	6 (3.64%)	4	2	0	3	3
Tetranucleotides	AAAT/ATTT	8 (4.85%)	7	1	0	5	3
	AATG/ATTC	1 (0.61%)	1	0	0	1	0
	AATT/AATT	2 (1.21%)	2	0	0	1	1
Pentanucleotides	AAAAT/ATTTT	1 (0.61%)	0	1	0	0	1
	AATGC/ATTGC	2 (1.21%)	0	0	2	2	0
Total		165 (100%)	110 (66.7%)	21 (12.7%)	34 (20.6%)	107 (64.8%)	58 (35.2%)

**Table 4 genes-13-01184-t004:** The long repeat sequences, including minisatellite sequences (M), forward repeat sequences (F), reverse repeat sequences (R), complementary repeat sequences (C), and palindromic repeat sequences (P), in the cp genome of *Q. litseoides*.

No.	Repeat Type	Repeat Length (bp)	Region	Location	No.	Repeat Type	Repeat Length (bp)	Region	Location
1	M	19	LSC	IGS (*trnF-GAA, ndhJ*)	25	R	33	LSC, LSC	*clpP*
2	M	20	IRa	*rpl12*	26	C	34	LSC, LSC	IGS (*rps16, trnQ-UUG*)
3	M	21	IRa	*ycf2*	27	C	30	LSC, LSC	IGS (*petA, psbJ*)
4	M	31	IRa	IGS (*rrn4.5S, rrn5S*)	28	P	56	SSC, SSC	IGS (*ndhD, psaC*)
5	M	31	IRb	IGS (*rrn5S, rrn4.5S*)	29	P	44	LSC, LSC	IGS (*psbT, psbN*)
6	M	21	IRb	*ycf2*	30	P	40	IRa, IRb	*rpl12*
7	M	20	IRb	*rpl12*	31	P	40	IRa, IRb	*rpl12*
8	F	40	IRa, IRa	*rpl2*	32	P	38	LSC, LSC	IGS (*atpF, atpH*)
9	F	40	IRb, IRb	*rpl2*	33	P	34	SSC, SSC	*ycf1*
10	F	39	LSC, IRa	*ycf3*	34	P	39	LSC, IRb	*ycf3*
11	F	40	IRa, SSC	IGS (*rps12, trnV-GAC*)	35	P	40	SSC, IRb	*ndhA*
12	F	30	IRa, IRa	IGS (*rrn4.5S, rrn5S*)	36	P	39	LSC, LSC	IGS (*trnT-GGU, psbD*)
13	F	30	IRb, IRb	IGS (*rrn5S, rrn4.5S*)	37	P	30	LSC, LSC	*trnS-GGA*
14	F	30	LSC, LSC	*psaB*	38	P	30	IRa, IRb	IGS (*rrn4.5S, rrn5S*)
15	F	30	LSC, IRa	*ycf3*	39	P	30	IRa, IRb	IGS (*rrn4.5S, rrn5S*)
16	F	30	IRa, SSC	IGS (*rps12, trnV-GAC*)	40	P	32	LSC, LSC	IGS (*trnH-GUG, psbA*)
17	F	30	IRa, IRb	*ycf1*	41	P	30	LSC, IRb	*ycf3*
18	F	32	IRa, IRa	*ycf2*	42	P	30	IRa, IRa	*ycf1*
19	F	32	IRb, IRb	*ycf2*	43	P	30	SSC, IRb	*ndhA*
20	F	30	LSC, LSC	*trnS-GGA*	44	P	30	IRb, IRb	*ycf1*
21	F	30	LSC, LSC	*trnG-GCC*	45	P	32	LSC, LSC	IGS (*rbcL, accD*)
22	R	31	LSC, LSC	IGS (*trnR-UCU, atpA*)	46	P	32	IRa, IRb	*ycf2*
23	R	31	LSC, LSC	*clpP*	47	P	32	IRa, IRb	*ycf2*
24	R	31	LSC, LSC	IGS (*atpA, atpF*)	48	P	30	LSC, LSC	*trnS-UGA*

## Data Availability

The data that support the finding of this study are openly available in the GenBank of NCBI at https://www.ncbi.nlm.nih.gov (accessed on 25 May 2022), reference number (ON598394).

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
