# Peer review of "Complete Chloroplast Genome of an Endangered Species Quercus litseoides, and Its Comparative, Evolutionary, and Phylogenetic Study with Other Quercus Section Cyclobalanopsis Species"

_genes, 2022, doi:10.3390/genes13071184_

Round 1

Reviewer 1 Report

The authors report the complete sequence of the Quercus litseoides chloroplast genome and the genomic comparison with other Quercus species. This work represents an important advance in genomic data generation on this species and its phylogenomic relationship with other species. However, there are some things that could be improved.

Lines 108-109: Possible typo error. Please, confirm that this sentence is correct: “…were set to two, seven, and seven, respectively”

Lines 155-157: Authors claim that the chloroplast genome has a quadripartite structure: LSC, SSC, and IR. However, this is not clearly seen in figure 1. I assume that the border of circle 3 tries to show this, but it is not clear. It is suggested that annotations be added to make this clear in Figure 1. Where start and end of each region?

Lines 280-281: How many homologous sequences were used to construct the phylogeny?

Lines 284-285: This sentence is confused “… the Quercus sections Quercus and Lobatae formed…” Please, check this sentence.

Figure S1: The description of figure S1 does not match with the keys shown in the figure, for example, it says that the cyan color represents introns, but in the figure's key, the cyan color indicates UTRs. What does CNS stand for? Authors should check the color's key and description of this figure.

In this same figure, it is not clear where the different regions (LSC, SSC, IR) mentioned in the text begin or end (242-248). It is suggested to indicate where these regions begin or end, something like in figure 3 on the X-axis.

Author Response

Responses to Reviewers

Manuscript ID genes-1770000

“Complete chloroplast genome of an endangered species Quercus litseoides, and its comparative, evolutionary, and phylogenetic study with other Quercus section Cyclobalanopsis species”

Our itemized answers and remarks are highlighted in blue.

COMMENTS FOR AUTHOR:

Reviewer #1:

Lines 108-109: Possible typo error. Please, confirm that this sentence is correct: “…were set to two, seven, and seven, respectively”

Answer:

Thank you. We have changed the sentence as bellow: “The alignment parameters for match, mismatch, and indels were set to be two, seven, and seven respectively.”. (Line 110‒111)

Lines 155-157: Authors claim that the chloroplast genome has a quadripartite structure: LSC, SSC, and IR. However, this is not clearly seen in figure 1. I assume that the border of circle 3 tries to show this, but it is not clear. It is suggested that annotations be added to make this clear in Figure 1. Where start and end of each region?

Answer:

Thank you for your suggestions. We have improved the Figure 1 to display the quadripartite structure of Quercus litseoides. (Figure 1)

Lines 280-281: How many homologous sequences were used to construct the phylogeny?

Answer:

Homblock was used to select homologous sequences for the phylogeny. For each species, the software will generate one sequence with 88,044 bp. Finally, a total of 37 homologous sequences were used to construct the phylogeny. (Line 285)

Lines 284-285: This sentence is confused “… the Quercus sections Quercus and Lobatae formed…” Please, check this sentence.

Answer:

Thank you. We have changed the sentence as bellow: “The results showed that the section Quercus and section Lobatae formed one clade.”. (Line 290)

According to the latest taxonomic systematics, Quercus was divided into two subgenera with eight sections in the genus. Subgenus Quercus consists of five sections: section Protobalanus, section Ponticae, section Virentes, section Quercus, and section Lobatae; while subgenus Cerris consists of three sections: section Cerris, section Ilex and section Cyclobalanopsis.

Figure S1: The description of figure S1 does not match with the keys shown in the figure, for example, it says that the cyan color represents introns, but in the figure's key, the cyan color indicates UTRs. What does CNS stand for? Authors should check the color's key and description of this figure.

In this same figure, it is not clear where the different regions (LSC, SSC, IR) mentioned in the text begin or end (242-248). It is suggested to indicate where these regions begin or end, something like in figure 3 on the X-axis.

Answer:

Thank you. We have modified the keys and description of Figure S1 according to your suggestion.

Different colors represent different genome regions: blue for protein-coding (exon), cyan for tRNA or rRNA, and red for conserved non-coding sequences (CNS). (Figure S1)

Reviewer 2 Report

The authors assembled and described the genome features, phylogenetics and evolution of Quercus litseoides. It is confirmed that the chloroplast genome of this species has never been reported and the finding of this study is valuable to the in-depth understanding of Quercus. Overall, the manuscript is well-written; there is still need for minor language proofing. However, based on my humble opinion, I would like to suggest few things to improve the quality of this manuscript.

Abstract: results should be presented in past tense.

L20 8 ribosomal > eight ribosomal

L28 positive selection pressure when compared to..?

L29 phylogeny > phylogenetic analysis

L43 is nested in

L43 has been largely investigated for their phylogenetic relationships

L59 is systematically belong to

L70 cheaper > affordable

L76 PCG? should it be CDS instead?

L130 For this section, you should be comparing your data with a particular species to show whether the CDSs are positively or negatively selected. But I cannot detect any information on that. Please justify.

L139 you do not use the cp genomes to reconstruct the trees, but you used cp genome sequences.

L148 two things: NJ is not a reliable tree analysis. I would strongly suggest to replace it with a Bayesian inference (BI) tree. Second, MEGA-X is not suitable to reconstruct trees that are derived from big data. Please opt to use RAxML or similar pipelines that are proven to be useful for big data analysis.

L154 clean reads? or raw reads?

L182 four functional categories

L184 there is no such thing as unknown genes. It is the function that is unknown.

L280 88,044

L360 The genus name should be spelled in full when appears as the first word in the sentence.

L373 The discussion of the molecular placement of Q. litseoides was not properly discussed in this section. The authors should should compare their finding to those using short gene sequences, or nuclear gene sequences, if available.

Figure 5: I suppose ML bootstrap support should be in form of %, not decimal points.

Author Response

Responses to Reviewers

Manuscript ID genes-1770000

“Complete chloroplast genome of an endangered species Quercus litseoides, and its comparative, evolutionary, and phylogenetic study with other Quercus section Cyclobalanopsis species”

Our itemized answers and remarks are highlighted in blue.

COMMENTS FOR AUTHOR:

Reviewer #2:

Abstract: results should be presented in past tense.

Answer:

Thank you for your suggestions. We have revised the abstract and highlighted the revised sentences in blue as below.

Abstract: Quercus litseoides, an endangered montane cloud forest species, is endemic to southern China. To understand the genomic features, phylogenetic relationships, and molecular evolution of Q. litseoides, the complete chloroplast (cp) genome was analyzed and compared in Quercus section Cyclobalanopsis. The cp genome of Q. litseoides was 160,782 bp in length, with an overall guanin and cytosine (GC) content of 36.9%. It contained 131 genes, including 86 protein-coding genes, eight ribosomal RNA genes, and 37 transfer RNA genes. A total of 165 simple sequence repeats (SSRs) and 48 long sequence repeats with A/T bias were identified in the Q. litseoides cp genome, which were mainly distributed in the large single copy region (LSC) and intergenic spacer regions. The Q. litseoides cp genome was similar in size, gene composition, and linearity of the structural region to those of Quercus species. The non-coding regions were more divergent than the coding regions, and the LSC region and small single copy region (SSC) were more divergent than the inverted repeat regions (IRs). Among the 13 divergent regions, 11 were in the LSC region and only two were in the SSC region. Moreover, the coding sequence (CDS) of the six protein-coding genes (rps12, matK, atpF, rpoC2, rpoC1, and ndhK) were subjected to positive selection pressure when pairwise comparison of 16 species of Quercus section Cyclobalanopsis. A close relationship between Q. litseoides and Quercus edithiae was found in the phylogenetic analysis of cp genomes. Our study provided highly effective molecular markers for subsequent phylogenetic analysis, species identification, and biogeographic analysis of Quercus.

L20 8 ribosomal > eight ribosomal

Answer:

Thank you. We have made the changes in the abstract above. (Line 20)

L28 positive selection pressure when compared to..?

Answer:

Thank you. We have made the changes in the abstract above. (Line 28-30)

L29 phylogeny > phylogenetic analysis

Answer:

Thank you. We have made the changes in the abstract above. (Line 31)

L43 is nested in

L43 has been largely investigated for their phylogenetic relationships

Answer:

Thank you for your suggestions. We have changed the sentence as bellow: “The taxonomy, which section Cyclobalanopsis is nested in Quercus, has been largely investigated for their phylogenetic relationships.”. (Line 45-46)

L59 is systematically belong to

Answer:

Thank you. We have changed the sentence as bellow: “Leaf epidermal features of Q. litseoides (uniseriate trichome with single-celled trichome base (STB)) showed that it is systematically belong to the STB group.”. (Line 60-61)

L70 cheaper > affordable

Answer:

Thank you. We have changed the sentence as bellow: “Owing to the benefits of next-generation sequencing technologies, whole cp genome sequencing is affordable and more efficient than ever before.”. (Line 72)

L76 PCG? should it be CDS instead?

Answer:

Thank you. The data used for positive selection analysis was the coding sequence (CDS) of the 78 shared protein-coding genes (PCGs), so the description related to selection pressure analysis has been modified. (Line 78, 134-139, 271-278)

L130 For this section, you should be comparing your data with a particular species to show whether the CDSs are positively or negatively selected. But I cannot detect any information on that. Please justify.

Answer:

Thank you. We extracted the CDS of the 78 shared PCGs of 16 Cyclobalanopsis species and calculated the corresponding Ka and Ks values after alignment. Based on each CDS, DNAsp software was used to calculate the Ka and Ks values by pairwise comparison among 16 species. Then, based on Jukes & Cantor model, the comprehensive value of Ka and Ks of each CDS was obtained for selection pressure analysis.

L139 you do not use the cp genomes to reconstruct the trees, but you used cp genome sequences.

Answer:

Thank you. We have changed the sentence as bellow: “Thirty-four complete cp genome sequences, comprising one new cp genome sequence and 33 cp genome sequences of five sections of Quercus species from the GenBank database, were used to reconstruct phylogenetic relationships.”. (Line 141-142)

L148 two things: NJ is not a reliable tree analysis. I would strongly suggest to replace it with a Bayesian inference (BI) tree. Second, MEGA-X is not suitable to reconstruct trees that are derived from big data. Please opt to use RAxML or similar pipelines that are proven to be useful for big data analysis.

Answer:

Thank you for your suggestions. We have reconstructed the Maximum Likelihood (ML) and Bayesian Inference (BI) trees using IQtree v.1.6.12 and MrBayes v3.2, respectively.

Accordingly, we revised the materials and methods and the results of the paper. (Line 149-156, 287-300)

L154 clean reads? or raw reads?

Answer:

Thank you. The raw reads were filtered by SOAPnuke v1.3.0 Filtering criteria were as follows: (1) remove reads with N base content exceeding 5%; (2) remove reads of low quality (mass value less than or equal to 5) with 50% base number; (3) remove reads contaminated by adapter. In this sequencing, the sequencing volume of raw data is five Gb, and the data volume of clean data is also five Gb after data filtering.

L182 four functional categories

Answer:

Thank you. We have changed the sentence as bellow: “Based on the cp genome annotation of Q. litseoides, 113 unique genes were divided into four functional categories with 18 groups.”. (Line 187)

L184 there is no such thing as unknown genes. It is the function that is unknown.

Answer:

Thank you. We have changed the sentence as bellow: “Among the 113 unique genes, there were 60 genes related to transcription and translation, 44 genes related to photosynthesis, five genes related to biosynthesis, and four genes whose functions were unknown.”. (Line 189)

L280 88,044

Answer:

Thank you. We have changed the sentence as bellow: “Thirty-seven homologous sequences with length of 88,044 bp were generated.”. (Line 285)

L360 The genus name should be spelled in full when appears as the first word in the sentence.

Answer:

Thank you for your suggestions. We have changed the sentence as bellow: “Quercus litseoides lives at high altitudes and must adapt to high UV radiation intensity, hypoxia, low temperature, and drought stress.”. (Line 367)

L373 The discussion of the molecular placement of Q. litseoides was not properly discussed in this section. The authors should compare their finding to those using short gene sequences, or nuclear gene sequences, if available.

Answer:

Thank you. There is no phylogenetic research contains Quercus litseoides before. Only morphological study has been done by Deng et al., 2014.

Figure 5: I suppose ML bootstrap support should be in form of %, not decimal points.

Answer:

Thank you. We have added new analysis and changed the figure totally. (Figure 5)